# Women Entrepreneurs Who Break through in Reward-Based Crowdfunding: The Influence of Entrepreneurial Orientation

**Ke Zhao** [1], **Hongwei Wang** [1,*] and **Wei Wang** [2]

1   School of Economics and Management, Tongji University, Shanghai 200092, China; 1910478@tongji.edu.cn
2   College of Business Administration, Huaqiao University, Quanzhou 362021, China; wayswang@gmail.com
*   Correspondence: hwwang@tongji.edu.cn; Tel.: +86-021-65984225

**Abstract:** Based upon social identity theory, having a strategic entrepreneurial orientation is crucial for ventures seeking funding, since entrepreneurial orientation (viewed as an entrepreneurial organizational culture) has a significant influence on investors' decision-making for resource allocation. However, the attitude of investors toward women entrepreneurs' behavioral orientation is controversial. Women entrepreneurs may not benefit from specific behavioral orientations because of gender bias. In this study, we had a sample of 5105 'design and technology' campaigns with explicit gender information on Kickstarter, utilizing a computer-aided text analysis dictionary of entrepreneurial orientation to examine whether the five dimensions of entrepreneurial orientation signals affected the relationship between women entrepreneurs and their performance in reward-based crowdfunding. Our findings demonstrated that investors tended to support women entrepreneurs who displayed many of the signals for autonomy and risk-taking, while backers were skeptical of women entrepreneurs displaying a high number of the signals for proactiveness. This study will enable an in-depth understanding of the link between investors' decision-making and women's entrepreneurial behaviors, in addition to determining which specific entrepreneurial behavior is important for helping women entrepreneurs to obtain funding in the context of reward-based crowdfunding, from a practical perspective.

**Keywords:** women entrepreneurs; entrepreneurial orientation; social identity theory; crowdfunding

## 1. Introduction

Striving for external funding is an important way for start-ups to achieve long-term development and rapid expansion [1]. Female entrepreneurs have more trouble than males do when seeking bank loans [2], institutional capital [3], and private equity [4]. In formal financing channels, male entrepreneurs typically receive more venture funding from professional investors than female entrepreneurs do [5,6], since an "entrepreneur is a man, not a woman" in a venture capitalist's ideal world [3]. Furthermore, entrepreneurship is stereotypically looked upon as a masculine trait [7], and investors are not instinctively connecting women to such behaviors [8]. In relation to gender, scholars have advocated that in order to obtain funds from traditional forms of capital, women entrepreneurs have had to reduce stereotypically feminine characteristics and disguise themselves as "men" or collaborate with a man [9,10].

As a cornerstone of the literature on organization-level entrepreneurship [11], entrepreneurial orientation (EO) is a corporate strategic posture that captures the particular practices and decision-making activities which lead enterprises to a new opportunity and create value [12,13]. The five main dimensions of EO (autonomy, competitive aggressiveness, innovativeness, proactiveness, and risk-taking) have been used for distinguishing and characterizing the main entrepreneurial processes [14]. Venture capitalists pay great attention to start-up ventures' EO due to its demonstrated potential to ensure their firm's continued sustainable growth and increase their firm's profitability [15,16]. Choosing an

appropriate EO is essential for early start-ups if they are to enhance a firm's innovation performance and produce long-term sustainable improvement in that performance [13], and thereby obtain venture capital or crowdfunding support. However, previous studies have demonstrated that, due to gender bias, women entrepreneurs generally have lower self-efficacies and behavioral orientations; for instance, they are reported to be more risk-averse [17], nonaggressive [9], and passive toward opportunity [18] when compared to men entrepreneurs in the entrepreneurial process, with such reports leading funders to perceive a behavioral and competency difference, and to discriminate against females [19].

Even though women entrepreneurs are disadvantaged in formal financing channels, informal fundraising environments may bring unexpected benefits to women [20]. In crowdfunding markets with asymmetric information, women-led enterprises have greater advantages than men-led enterprises due to trustworthiness of their judgments [20]. Women business owners are considered to be more reliable than their male counterparts; meanwhile, crowd-funders may feel more fulfilled and motivated to fund projects led by women. However, limited research has explained how entrepreneurs' behavioral orientation impacts the relationship between female entrepreneurs and crowdfunding performance in the emerging fundraising context. According to social identity theory [21], a social identity is "the individual's knowledge that he belongs to certain social groups together with some emotional and value significance to him of this group membership" [22]. The social identity concept (which comprises the evaluative, emotional, and other social psychological correlates of in-group classification) impacts how an entrepreneur self-categorizes, and thus influences the entrepreneur's behavioral orientation [23,24].

Our study builds on social identity theory and develops a moderation model which considers the role of EO in fundraising decisions. In this study, we selected a sample of 5105 'design and technology' campaigns with explicit gender information on Kickstarter, and utilized a computer-aided text analysis (CATA) dictionary of EO to explore the underlying psychological mechanism of investors who generate a "gender gap" in crowdfunding decisions, while also examining whether the five dimensions of entrepreneurial behavioral orientation derived from social identity theory are a source of advantage for women entrepreneurs' reward-based crowdfunding performance, measured using the total amount of money raised, the total number of backers who support the crowdfunding project, and crowdfunding success. The measure of entrepreneurial orientation used in the current study is based on five EO dimensions: autonomy, competitive aggressiveness, innovativeness, proactiveness, and risk-taking [14]. Our results imply that, even though women entrepreneurs might get a negative effect from presenting high signals of proactiveness, enhancing signals of risk-taking and autonomy may have a beneficial impact on the fundraising outcomes of women-led enterprises in the reward-based crowdfunding market. By identifying linguistic features of online project narratives, we reappraise the common negative views of women entrepreneurs as disadvantaged both in their accessing of capital and resources, as well as in the representation of their EO signals [25].

Our current study makes three major contributions to the entrepreneurship, social identity theory, and women entrepreneurial sustainability literature. First, we seek the driving EO signals behind the funding advantages for women entrepreneurs on online crowdfunding platforms based upon social identity theory. Although prior studies have suggested that women entrepreneurs have certain advantages in online crowdfunding, little is known about the contribution of EO to the gender gap [26]. As our results demonstrated, in reward-based crowdfunding, women entrepreneurs presenting signals of a high level of autonomy and risk-taking are rewarded, whereas releasing strong signals of proactiveness would run counter to this. This study enables an in-depth understanding of the link between investment decision making and judgments on women's entrepreneurial behaviors [27,28], as well as testing whether the social identity concept-derived EO mechanism is a source of advantage for women entrepreneurs.

Second, this study hopes to contribute to the social identity literature by exploring entrepreneurs' behavioral orientations in crowdfunding, and respond to the call of gender

scholars to examine heterogeneity among female and male entrepreneurs [29,30]. From the perspective of entrepreneurial behavioral orientations, our study not only goes beyond demographics to reveal the social psychological mechanisms that affects the judgment of investors, but also investigates the five dimensions of EO in different genders in online crowdfunding contexts [31].

Third, while prior studies has emphasized critical issues surrounding the financing of women-led ventures [20], there is little theory-driven research on this subject. Here, our analysis offers two crucial findings for further investigation. First, as the factors influencing backers' judgement in the two situations could be different, it is not always easy to simply adapt common entrepreneurial financial notions from the funding of men-led microenterprises into the setting of women-led microenterprises. Second, the importance of entrepreneurial behavior is then illustrated in relation to women entrepreneurs, a situation where the stakeholders' expectations are ambiguous. We highlight that women entrepreneurs have to move beyond trust if they are to attract potential investors who take advantage of the entrepreneurial orientations of autonomy and risk-taking.

The rest of the paper is structured as follows: In Section 2, we describe the theoretical background, including the notion of social identity theory, EO constructs, women entrepreneurs, and crowdfunding performance. We then develop hypotheses. Section 3 describes the sample and introduces the research methodology. Section 4 presents a detailed analysis of the results. Section 5 discusses the results in detail and concludes with a general discussion of the theoretical and practical implications of the results.

## 2. Theory and Hypotheses

### 2.1. Social Identity Theory in Entrepreneurship

Social identity theory originates in the literature on social psychology, and comprises three fundamental processes: categorization, identification, and comparison [21,22]. The first component, 'categorization', refers to people grouping themselves into a community; 'identification' is the conviction that one possesses the common features of the members of the group; 'comparison' is the process of assessing the virtues, standing, and prestige of one's group in relation to other groups [22,32]. "Because firm creation is an inherently social activity, and organizations are themselves social constructions" [33], social identity theory explains how the self-perception of entrepreneurs influences the methods and results of firm formation and interprets why founder firms might be entrepreneurial [23,34,35]. Many business and management activities (such as discovering different markets, fundraising, establishing a new company, staffing executives, and negotiating with stakeholders) could enact and remind an entrepreneur of his or her entrepreneurial social identity [36].

Social identity mainly comes from group membership or qualification, and sharing important traits with role models in an elite group may encourage the development of an entrepreneurial social identity, reinforcing self-efficacy beliefs [36]. Entrepreneurs work to develop or maintain a positive social identity, such as a strong entrepreneurial orientation or characteristic, in order to boost their self-esteem. This positive social identity mostly results from favorable comparisons between the in-group and related out-group [22]. Social identity researchers have found that a greater commitment to the group is led by in-group identification, and that people who identify with the group are strongly attracted to the group [37,38]. Entrepreneurs may be inspired to emulate other successful or entrepreneurial models, and they may also operate in accordance with group customs and values [39]. Being influenced by the interactions with venture capitalists and investors, entrepreneurs focus on attaining high financial performance; they are, in essence, business builders, risk takers, and innovators [40]. These aspects of entrepreneurship correspond closely to the five key EO dimensions, namely, autonomy, competitive aggressiveness, proactiveness, innovation, and risk-taking [36].

*2.2. EO and Crowdfunding Performance*

Signaling theory proposes that actors in asymmetrical information relationships signal information to work together, in addition to stating how signaling systems affect funding performance [41,42]. In the crowdfunding context, information asymmetry would be a problem between ventures and investors since project creators are limited in the amount of information they can provide, and structure the online representation of their projects to potential investors [43,44]. Nevertheless, the descriptions of crowdfunding campaigns serve as vital information conduits for a project. On reward-based crowdfunding platforms, the signaling system consists of campaign creators, potential investors, and the entrepreneurial orientation signals that are associated with campaign creators' entrepreneurial orientation behavior, allowing potential investors to select which crowdfunding project to back. Signaling information using crowdfunding rhetoric has been used to investigate the effects of entrepreneurial orientation [45], Machiavellianism [46], and narcissism [41]. Hence, investors can still investigate ventures for signals of behavioral orientation and quality when making a decision, and having a strategic entrepreneurial orientation is crucial for ventures seeking funds [47–49].

An EO "leads a firm and its members to constantly search and filter information for new product ideas and process innovations that will lead to greater profitability" [50]. It is well acknowledged that entrepreneurship and business performance are related [51–53], and as a focal point of entrepreneurship literature, an EO is measured as both a behavioral construct and a tractive force behind the organizational pursuit of entrepreneurial activities [54,55]. All the EO dimensions (including a willingness to take risks and innovate, a tendency toward autonomy, a propensity to be proactive to marketplace opportunities, and aggression relative to competitors) have been presented in discourse published by business ventures [56,57].

It has been extensively acknowledged that EO, when viewed as an entrepreneurial organizational culture [51], has a significant influence on investors' decision-making for resource allocation [58,59]. In debt-based crowdfunding, Moss et al. (2015) [59] revealed that while signals of courage, conscientiousness, warmth, and empathy have a negative impact on microlending performance, signals of autonomy, risk-taking, and competitive aggressiveness positively impact microlending performance. For reward-based crowdfunding, Gc and As (2020) [55] explored the signal intensity of EO dimensions and found that there is an inverted-U-shape relationship between signals of autonomy, innovativeness, competitive aggressiveness, risk-taking and crowdfunding support, whereas there is a positive non-monotonic relationship between proactiveness signals and crowdfunding support. Furthermore, the potential funders' difference between reward-based crowdfunding platforms and traditional sources of financing is that crowd funders are more likely to be driven by a desire to help others, rather than aiming at a direct financial return. Thus, the signals of behavioral orientations released by a venture are more likely to grab backers' attention [60,61].

*2.3. Baseline Relationship between Women Entrepreneurs and Crowdfunding Performance*

With the rapid development of emerging fundraising channels, the fundraising performance of women entrepreneurs has also begun to attract attention [62,63]. Studies have found that being affected by the gender homophily, linguistic style, and trustworthy, women entrepreneurs are expected to obtain higher funding amounts and a larger number of backers than men entrepreneurs in crowdfunding [20,64,65]. Regarding gender issues on crowdfunding platforms, huge attention has been given to equity crowdfunding. For example, Battaglia et al. (2021) found that women-led firms are more successful in receiving equity crowdfunding than men-led firms, and that benefit is stronger in societies where women have fewer chances [66]. Bapna and Ganco (2020) confirmed that the gender disparities present in traditional equity financing are mitigated in the establishment of equity crowdfunding [67]; however, Malaga et al. (2018) offer factual proof that the democratization of access to financing for woman-owned enterprises has been hardly affected by

equity crowdfunding [68]. As for gender-related differences in the behavior of investors in equity crowdfunding, Mohammadi and Shafi (2018) show that female investors are more risk-averse than male investors [69].

Investment decisions made through equity crowdfunding may result in a potential return on investment, while reward-based crowdfunding is described as a "way of giving back to a world in which there is so much negativity" [70] for supporters, who are open about how social norms affect their fundraising choices. Compared to equity crowd-funding platforms, backers on reward-based crowdfunding platforms such as Kickstarter may feel more fulfilled and motivated to fund projects led by women than those led by men [20] since the platform is built upon shared norms of community, cooperation, and giving [71]. The desire to experience the positive emotional state that results from helping others in need motivates the warm glow charitable giving [72]; for instance, in reward-based crowdfunding, crowd funders reward themselves with a 'warm glow feeling' when helping others. Similarly, if funders increase their utility by investing in a campaign that might advance gender equality, then it is reasonable that they would be eager to invest in women-led businesses [73]. As a result, crowd funders seem more willing to back women entrepreneurs.

We investigate this underlying main assumption to find whether women entrepreneurs do have advantages in the reward-based crowdfunding context. We hypothesize:

**H1:** *In reward-based crowdfunding, women entrepreneurs are more likely than men to obtain financial backing.*

### 2.4. The Moderating Effects of the Entrepreneur Orientation
2.4.1. Autonomy

As an important component of an EO, Autonomy enables organizational members to seek new opportunities and advantages without being constrained by extant norms or organizational structures, which affords them the freedom and flexibility to foster new ideas and develop entrepreneurial initiatives [74–76]. Ventures which are highly autonomous are independent and self-determined. The effective use of autonomy by ventures is necessary to transform existing organizations by updating their strategic capabilities [75] and redeploying resources [77]. Prevailing research supports the view that ventures may suffer financial problems if they are overly dependent and require agreement to be reached before making a decision to launch an entrepreneurial initiative [78]. In contrast, EO related autonomy encourages organizations' innovation and transformation and increases their competitiveness and performance [79]. Reward-based crowdfunding platforms such as Kickstarter provide a place where creative campaigns can gain financial support from crowd funders. Autonomy has a crucial role in the success of crowdfunding because the main goal of the platform is to strengthen the independence of small businesses [55,75,80].

According to descriptive gender stereotypes, in contrast to stereotype of women, which emphasize obedience, respect, and self-effacing behavior, the stereotype of men assumes their being characterized by independence, self-reliance, and decisiveness [81,82]. Because independence is often interpreted as being masculine, women's development of a need for autonomy might be being restricted. Despite knowing this, based upon the Lack of Fit model [82,83], the perceived mismatch between feminine traits and masculine-typed job requirements can lead to unfavorable expectations about women's likely success, which can have serious repercussions on how women's performances are assessed and handled at work. In formal financing channel, women-led enterprises may confront more difficulties than men-led enterprises due to gender bias; nevertheless, women entrepreneurs have certain advantages in online crowdfunding contexts [64,73]. Because of the information asymmetry of online crowdfunding and the implicit gender bias of investors, entrepreneur women are seen as being more reliable than their male counterparts [20], something which could largely improve women entrepreneurs' confidence and independence. From the perspective of crowd funders, they would be willing to see and support a women

entrepreneur who has a high degree of autonomy to successfully realize an idea based on entrepreneurial social identity theory [36]. Therefore:

**H2a.** *The relationship between a woman entrepreneur and reward-based crowdfunding performance is stronger when the entrepreneur has a high degree of autonomy than when she does not.*

### 2.4.2. Competitive Aggressiveness

The second dimension of EO is competitive aggressiveness, which is characterized as a variety of offensive tactics or aggressive reactions undertaken by companies that are heading for a strong competitive position in the marketplace [84]. It is significant for enterprises to compete aggressively for scarce resources, and especially for microenterprises that have less power than incumbents but need to survive and succeed [76]. Signals of aggressively pursuing the competition also imply that ventures could perform well by exploiting opportunities to provide novel types of value [76] and heighten the likelihood of resource acquisition [59]. Aggressiveness is not meant in regard to the elimination of competitors, but rather to achieve success and progress for the community [85]. As of result, microenterprises that assume an aggressive stance are thought to outperform others that are less competitive and less aggressive [59].

According to descriptive gender stereotypes, men are stereotypically defined as being powerful, ambitious, and competitive, whereas women's stereotypes have been defined by being kind, caring, and compassionate [81,82]. However, in the context of reward-based crowdfunding, women entrepreneurs' self-efficacy and awareness of competition tend to be raised, owing to their perception of being more trustworthy by backers. Likewise, crowd funders would value and support a women entrepreneur who has a high degree of competitive aggressiveness to directly challenge their rivals and strengthen relative competitive position by implementing a unique, diverse, and sustained series of competitive activities based on entrepreneurial social identity theory [36,86,87]. Therefore:

**H2b.** *The relationship between a woman entrepreneur and reward-based crowdfunding performance is stronger when the entrepreneur has a high degree of competitive aggressiveness than when she does not.*

### 2.4.3. Innovativeness

Another important aspect of EO is innovativeness, which is the act of coming up with fresh ideas, novel methods, and improved products. Typically, innovation involves two stages: idea generation and subsequent implementation [88,89]. It has become a distinct competitive advantage for microenterprises to have an orientation toward innovation as they are devoted to changing the present status of industry by generating novel integration of resources or by exploiting a new business model [59]. Besides, for microenterprises, continuous reinvention is seen as the key to organizational performance and longer-term development, in addition to having a vital impact on entrepreneurial success and the efficient use of resources [90,91]. As a consequence, microenterprises with a propensity for innovation are supposed to outperform other (less innovative) microbusinesses.

According to descriptive gender stereotypes, stereotypes of men have been defined by the supposition of their being creative, imaginative, and inventive, while women's stereotypes are defined by their being thought to be conforming, proper, and changeless in behavior [81,82]. Nevertheless, in the setting of reward-based crowdfunding, women entrepreneurs are more likely to enhance their creativity and innovativeness, as it is perceived as being more trustworthy. Crowd funders would appreciate and support a women entrepreneur who displays a high signal intensity of innovativeness to act more entrepreneurially and constantly experiment with new ideas for service and production in a rapidly changing environment and shortened production life cycles, based on entrepreneurial social identity theory [36,59,92]. Therefore:

**H2c.** *The relationship between a woman entrepreneur and reward-based crowdfunding performance is stronger when the entrepreneur has a high degree of innovation than when she does not.*

### 2.4.4. Proactiveness

As an unique element of the entrepreneurial process, the EO dimension of proactiveness refers to an "opportunity seeking, forward-looking perspective involving introducing new products or services ahead of the competition and acting in anticipation of future demand to create change and shape the environment" [93]. In a dynamic environment, the effective utilization of proactive strategies could help firms to seize market opportunities and gain a competitive advantage [94]. However, in comparison to less proactive ventures, proactive enterprises are seen as industry leaders and suggest a more competitive and profitable venture to backers. Accordingly, in crowdfunding, it is more beneficial for microenterprises to be proactive 'first movers' in responding to changing circumstances [41,93].

According to descriptive gender stereotypes, the distinguishing characteristics of the stereotypical man are seen as being positive, active, and receptive, while the stereotypical woman is seen as being passive, inactive, and driven [81,82]. Nevertheless, in the context of reward-based crowdfunding, women entrepreneurs are more likely to enhance their proactiveness, as it is perceived as being more trustworthy by funders. Crowd funders would encourage and support a women entrepreneur who displays a high signal intensity for proactiveness to take the initiative and anticipate market changes and future requirements, and influence the environment by updating techniques, services, and business strategies based on entrepreneurial social identity theory [36,76,95]. Therefore:

**H2d.** *The relationship between a woman entrepreneur and reward-based crowdfunding performance is stronger when the entrepreneur is highly proactive than when she is not.*

### 2.4.5. Risk-Taking

Finally, as a quality of entrepreneurial behavior and a quintessentially entrepreneurial characteristic, risk-taking indicates the "degree to which managers are willing to make large and risky resource commitments—for example, those which have a reasonable chance of costly failure" [76,96], offering the probability for great payoffs in munificent environments owing to the increased availability of resources in those environments. In crowdfunding, calculated risk-taking is a necessary and essential entrepreneurial capability for microenterprises, and ventures that display a risk-taking propensity have a desire to commit significant resources to a business campaign in spite of the difficulties involved in making new things [59]. It thus stands to reason that risk-taking microenterprises will outperform those that appear risk averse [59].

According to descriptive gender stereotypes, stereotypes of men have been defined by their perceived risk-taking, audacity, and boldness, whereas stereotypes of women have been defined by their perceived risk-aversion, conservatism, and care [81,82]. Nevertheless, in the setting of reward-based crowdfunding, women entrepreneurs are more likely to present a propensity of risk-taking, as it is perceived as being more trustworthy by funders. Crowd funders would be in favor of a women entrepreneur who displays a high signal intensity of risk-taking to launch bold actions in the face of uncertainty, and to view uncertainties as "opportunities" even if others perceive them as having little potential, based on entrepreneurial social identity theory [36,97]. Therefore:

**H2e.** *The relationship between a woman entrepreneur and reward-based crowdfunding performance is stronger when the entrepreneur is highly risk-taking than when she is not.*

Based on the hypotheses above, Figure 1 summarizes our research model.

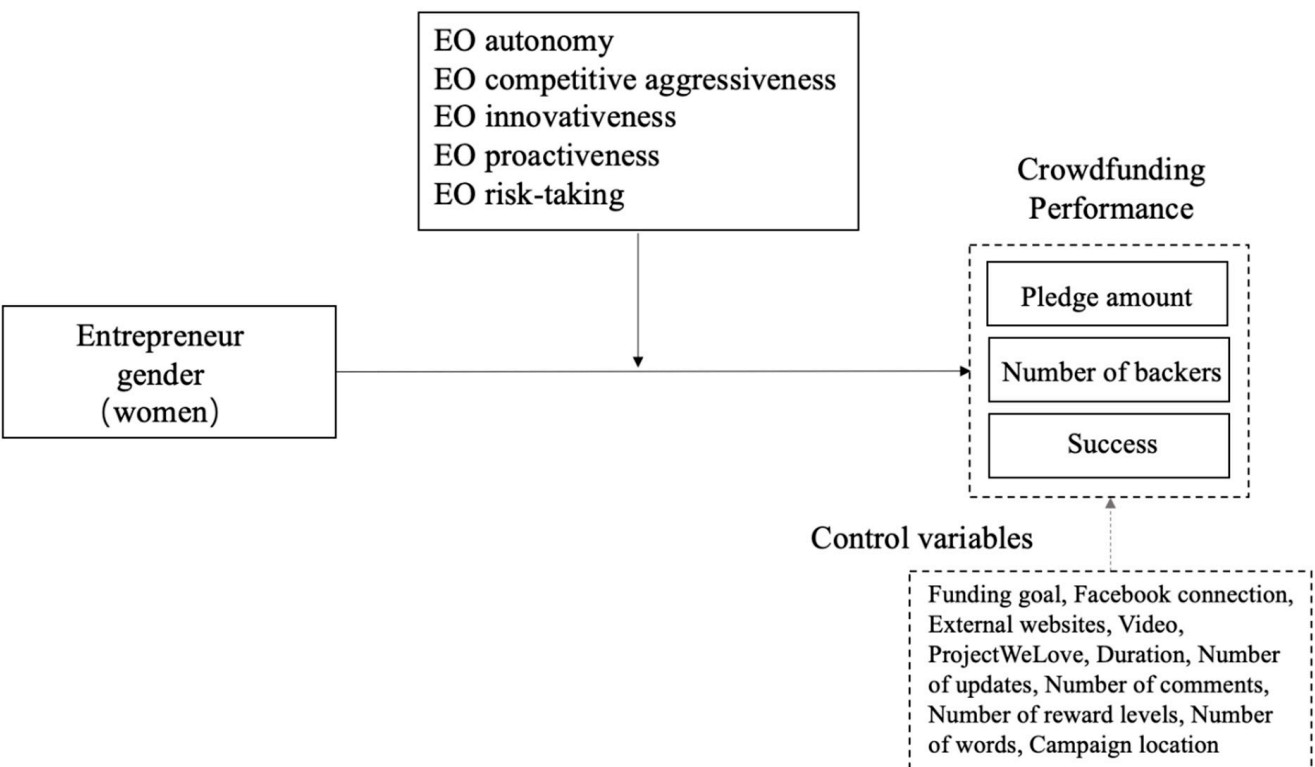

**Figure 1.** Overview of research model.

### 3. Data and Methodology

#### 3.1. Sample

Crowdfunding's primary purpose is to meet the capital needs of start-ups and small enterprises. The word "crowdfunding" refers to a very broad range of methods used by entrepreneurs and small enterprises to obtain money. According to crowdfunding's purpose (and what the supporters or investors get in return for the funding they provide to the entrepreneurs), there are four different types of crowdfunding projects that are typically recognized: loan-based, equity-based, donation-based, and reward-based [80]. Loan-based crowdfunding is the most similar to small company lending. Because they want to fund a loan with the highest interest rate at the lowest risk, investors' motives are entirely financial.

As a relatively new form of crowdfunding in the USA, equity-based crowdfunding is protected by the JOBS Act. Startups and small businesses could seek equity investments using this type of crowdfunding without having to submit complicated regulatory paperwork. Investors gain equity in the business along with the potential benefits, as well as risks that come with such investments.

Crowdfunding that is based on donations gives people or charities in need a platform to ask for and get donations from individuals. Donations are solely charitable acts, and donors have no return.

Reward-based crowdfunding, as it is explored in this paper, differs from the other three types of crowdfunding. The best example of reward-based crowdfunding is Kickstarter, which is both the most popular and the largest crowdfunding platform in the United States, raising over $6.8 billion since its inception, with more than 220,000 successful fundraising cases (See https://www.kickstarter.com/help/stats). It is a reward-based crowdfunding platform where supporters receive tangible rather than financial rewards for their financial contributions, and the platform adopts an all-or-nothing fundraising mechanism, meaning that projects can only be funded if they reach or surpass their funding target. As Kickstarter is described as a "way of giving back to a world in which there is so much negativity" [70] by supporters (who are open about how social norms affect their fundraising choices), we

choose this platform to explore the moderate effects of EO on the relationship between an entrepreneur's gender and reward-based crowdfunding performance.

To test the formulated hypotheses, we gathered information on reward-based crowdfunding projects through the kickstarter.com site. In particular, we first downloaded the dataset of Kickstarter from Web Robots website, which provided a scraper robot that crawled Kickstarter projects and collected data such as the project's name, pledge amount, number of backers, status, creator's name, duration, location, category, and funding goal. For other project information that are not included in the dataset (for instance, detailed project description, Facebook connection, external websites, video, number of updates, number of comments, number of reward levels and number of words), we utilized the Python program to crawl through the Kickstarter platform. Our sample of design and technology-related Kickstarter projects came from the period between the platform's introduction in 2009 and April 2022. A total of 100,090 initiatives made up our original sample, and 31,543 of those were successful and raised $2.54 billion in investment. Projects from design and technology category have a greater likelihood of providing backers with tangible goods and be entrepreneurial in nature compared to other categories in Kickstarter [55,71].

Given that each project has an equal chance of being chosen, random sampling maximizes the population's representation [98]. We then randomly selected 7000 initiatives launched on Kickstarter and coded them according to the gender of the creator. Particularly, we looked at the names, images, profiles, and attached videos of the creators. If a project was established by a group or had ambiguous gender information, it was not included. Eventually, we had a sample of 5105 campaigns with explicit gender information.

*3.2. Measures*

3.2.1. Dependent Variables

In this research, we measured reward-based crowdfunding performance along three major manifestations: pledged amount, number of backers, and success. The total amount pledged represents the amount of funding the crowdfunding project has received, denominated in US dollars. To code this variable, we converted other currencies to US dollars. Furthermore, since this is a highly skewed variable, we used a common (base 10) logarithmic transformation to explain the skewness of this variable. The transformed value, which may be higher or lower than the project's funding goal, is one of the most commonly used measurements of crowdfunding performance [80]. As one of the important indicators to measure the performance of crowdfunding, the number of backers represents the number of supporters participating in crowdfunding activities. This variable has been used in previous crowdfunding studies [80,99]. Owing to the large number of backers, they were logarithmically transformed in the regression to avoid bias caused by extreme values. We also examined the crowdfunding project's success, as Kickstarter operates on an all-or-nothing basis. Success was determined by whether or not the financing target was attained. When the pledge amount of the project exceeded the funding goal, it was regarded as a successful funding, and assigned a value of 1; otherwise, it was assigned 0.

3.2.2. Independent Variable

If a female entrepreneur launches a crowdfunding campaign, the dichotomous variable *entrepreneur gender* took on the value of 1; otherwise, it had a value of 0. We first looked at the profile name and description of the project creator in order to code this variable. We then looked to entrepreneur photographs, or attached videos if they were unclear as to gender. If the entrepreneur's gender was made explicitly obvious in the images or videos, they were coded.

### 3.2.3. Moderate Variables

Entrepreneurship scholars assert that text analysis can improve a construct's validity by allowing the construct to be objectively tested at the organizational level, increasing access to decision-making for individual founders who respond poorly to survey methodologies [100]. For this study, we relied on computer-aided text analysis (CATA) to capture EO signals of crowdfunding campaigns based on Kickstarter description pages, including the backstory of small businesses, project risks, funding purposes and aspirations for the future. CATA is a form of content analysis that measures structures by processing text into quantitative data based on the frequency of words. CATA was a good fit for our investigation because of its ability to analyze written or oral text verbatim and group words into predefined language categories; furthermore, it revealed some language styles that could have been easily overlooked by the human eye by providing relatively stable coding schemes [101]. LIWC (Linguistic Inquiry and Word Count) is one of the most commonly used linguistic analysis tools in academic research using CATA. It is often used to investigate the relationship between psychological variables and word use.

To measure the EO dimensions of entrepreneurs, we made use of word dictionaries that had already been created for the EO construct based on the Kickstarter platform [55]. Following the advice of Short et al. (2010) [57], Gc and As (2020) [55] used an inductive methodology to modify the word lists within the setting of reward-based crowdfunding, and eventually refine and supplement the original word lists of EO dimensions. Accordingly, it was appropriate for our study. Meanwhile, the dictionaries have also been used by other scholars to study the influence of EO on organization management and development [102–104]. Moreover, with the help of two experts who are familiar with the reward-based crowdfunding literatures, we examined the validity of five-word lists. The experts were informed of the meaning of the five dimensions of entrepreneurial orientation used in this study, and were asked to independently evaluate the meanings of these terms in the specific context of crowdfunding [55]. The results confirm the intercoder reliability of these structures (Krippendorff's $\alpha_{EO\text{-}autonomy}$ = 0.807; Krippendorff's $\alpha_{EO\text{-}competitive\ aggressiveness}$ = 0.824; Krippendorff's $\alpha_{EO\text{-}innovativeness}$ = 0.922; Krippendorff's $\alpha_{EO\text{-}proactiveness}$ = 0.901; Krippendorff's $\alpha_{EO\text{-}risk\text{-}taking}$ = 0.863), and Krippendorff's alpha value was higher than the conventionally accepted 0.8 consistent threshold [105,106]. Intercoder reliability has been used in several entrepreneurship studies [107,108].

We chose to use LIWC2015 software to measure the degree to which EO dimensions are emphasized in crowdfunding project backstories, so as to obtain the difference between what each enterprise is doing in its entrepreneurial behaviors. The narrative's length is controlled by a defined word count in the LIWC output; to be more precise, LIWC calculates the total number of words in the dictionary, divides that number by the total number of words in the entire article, then scales that result to a normalized value for every 100 words.

### 3.2.4. Control Variables

We controlled for a variety of factors that prior research on entrepreneur orientation had shown or argued would affect reward-based crowdfunding performance. In particular, we controlled for the amount of funding goal, because too high or too low of a project financing target may reduce the project's chance of success. We also performed a logarithmic transformation of this variable and measured it by taking logarithmic values. We also controlled for the duration of the project because the longer it lasts, the longer it will take creators to gather support for their initiatives [80]. A large number of reward levels provides backers with more product breadth and more choices [109]; thus, we controlled for the number of reward levels, which might increase the chance of receiving funding. We controlled for the number of project updates and the number of project comments, since these two variables present the level of detail in the crowdfunding campaign description [99]. In addition, some project creators often launched a functional website or an introductory video to promote the company's development history, as well as a series of derivative products and services. Thus, we controlled for the presence of external website

links and introductory videos [99] and added them as a binary variable. Moreover, we took into account the campaign's word count as a measure of the startup teams' level of effort and readiness [110]. We included a binary variable and controlled for the existence of Facebook connections as a proxy for the social networks of entrepreneurs. Furthermore, several crowdfunding campaigns were rated by the website as 'ProjectWeLove', which manifests the quality and acceptability of the project [20], so we controlled for this factor and treated it as a binary variable. Finally, we dummy coded the campaigns' location to control for their geographic origin because female entrepreneurs may be more likely to come from certain countries or may have a better chance of success if they do [20]. Table 1 provides a summary of variables.

**Table 1.** A summary of variables.

| Category | Variable | Description |
| --- | --- | --- |
| Dependent variables | Pledge amount<br>Number of Backers<br>Success | Pledge amount in U.S. dollars<br>Backers count<br>Pledge status |
| Independent variable | Entrepreneur gender | Women entrepreneur |
| Moderate variables | EO autonomy<br>EO competitive aggressiveness<br>EO innovativeness<br>EO proactiveness<br>EO risk-taking | Signal intensity of EO autonomy<br>Signal intensity of EO competitive aggressiveness<br>Signal intensity of EO innovativeness<br>Signal intensity of EO proactiveness<br>Signal intensity of EO risk-taking |
| Control variables | Facebook connection<br>External websites<br>Video<br>ProjectWeLove<br>Duration<br>Number of updates<br>Number of comments<br>Number of reward levels<br>Number of words<br>Funding goal<br>Geography: USA<br>Geography: Canada<br>Geography: Latin America<br>Geography: Europe<br>Geography: Asia<br>Geography: Oceania | Is a Facebook connection presented?<br>Is any external website presented?<br>Is an introduction video presented?<br>Is it a ProjectWeLove?<br>Pledge duration day<br>Number of updates<br>Number of comments<br>Number of pledge levels<br>Number of words used<br>Funding goal ($)<br>Geography: USA<br>Geography: Canada<br>Geography: Latin America<br>Geography: Europe<br>Geography: Asia<br>Geography: Oceania |

## 4. Results

### 4.1. Descriptive Statistics and Correlations

First, we performed a preliminary examination, comparing the means of crowdfunding performance and campaign characteristics by gender. We conducted a one-way ANOVA to compare between the means of these variables in woman-led campaigns and in man-led campaigns. As displayed in Table 2, woman-led campaigns had significantly higher crowdfunding performance from reward-based crowdfunding platforms (pledge amount: $26,331.46 vs. $18,372.94, $p < 0.10$; number of backers: 358.03 vs. 161.47, $p < 0.001$; success: 0.39 vs. 0.24, $p < 0.001$). In terms of EO signals, the mean of woman-led campaigns was even larger than the mean of man-led campaigns for autonomy, competitive aggressiveness, innovativeness, proactiveness, and risk-taking.

**Table 2.** Crowdfunding performance and campaign characteristics by gender.

| Variable | The Mean of Woman-Led Campaigns | The Mean of Man-Led Campaigns | *F*-Statistic [a] |
|---|---|---|---|
| Pledge amount ($) | 26,331.46 | 18,372.94 | 3.18 [+] |
| Number of Backers | 358.03 | 161.47 | 22.50 *** |
| Success | 0.39 | 0.24 | 69.00 *** |
| EO autonomy | 0.18 | 0.09 | 69.53 *** |
| EO competitive aggressiveness | 0.20 | 0.11 | 50.96 *** |
| EO innovativeness | 0.59 | 0.44 | 41.25 *** |
| EO proactiveness | 1.55 | 1.13 | 133.63 *** |
| EO risk-taking | 0.16 | 0.07 | 118.85 *** |
| Facebook connection | 0.30 | 0.37 | 11.79 ** |
| External websites | 0.73 | 0.74 | 0.14 [ns] |
| Video | 0.80 | 0.81 | 0.18 [ns] |
| ProjectWeLove | 0.13 | 0.09 | 10.87 ** |
| Duration | 35.76 | 35.92 | 0.10 [ns] |
| Number of updates | 6.10 | 4.60 | 16.83 *** |
| Number of comments | 83.55 | 47.52 | 5.47 * |
| Number of reward levels | 6.79 | 6.44 | 3.95 * |
| Number of words | 715.61 | 745.64 | 1.33 [ns] |
| Funding goal ($) | 55,177.77 | 69,669.57 | 0.29 [ns] |
| Geography: USA | 0.70 | 0.68 | 1.64 [ns] |
| Geography: Canada | 0.04 | 0.05 | 3.00 [+] |
| Geography: Latin America | 0.00 | 0.01 | 2.64 [ns] |
| Geography: Europe | 0.16 | 0.20 | 6.85 ** |
| Geography: Asia | 0.05 | 0.02 | 17.76 *** |
| Geography: Oceania | 0.03 | 0.03 | 0.00 [ns] |

Note. N = 5105. [a] *F*-statistic of one-way ANOVA test for significant differences between woman-led campaigns and man-led campaigns. [ns] $p > 0.10$, [+] $p < 0.10$, * $p < 0.05$, ** $p < 0.01$, *** $p < 0.001$.

Second, we examined the correlations. As shown in Table 3, the mean number of pledge amount, number of backers, and success were $9223.37, 187.81, and 0.26, respectively, and about 13 percent of the sample were woman-led campaigns. We also find preliminary evidence that the number of backers was positively correlated with pledge amount. The number of was significant, and strongly correlated with both pledge amount ($r = 0.82$; $p < 0.01$) and the number of backers ($r = 0.56$; $p < 0.01$).

### 4.2. Regression Analysis

For testing the hypotheses, we used SPSS software, building three models (model 1: OLS; model 2: negative binomial model; and model 3: Binary logistic model) for three dependent variables. Specifically, we ran a simple pooled-regression model for the continuous dependent variable of 'pledge amount', a generalized linear model for the continuous dependent variable of 'number of backers', and a binary logistic regression model for the dichotomous dependent variable of 'success'. Model 1 was a simple pooled-regression model estimated with OLS, which assumed that all possible correlations between the EO variables and the error term were captured by the set of control variables [55]. Model 2 was a generalized linear model that considered the distribution of the dependent variable by using a negative binomial distribution and a link function that connected the linear prediction to the result. Because the empirical standard deviation is much greater than the mean, a negative binomial model is preferred and often used for counting outcomes. Model 3 was a Binary logistic regression model, which have become increasingly common, as a dependent variable in such an analysis is dichotomous.

**Table 3.** Descriptive statistics and correlations.

| | Variable | Mean | S.D. | 1 | 2 | 3 | 4 | 5 | 6 | 7 | 8 | 9 | 10 | 11 | 12 |
|---|---|---|---|---|---|---|---|---|---|---|---|---|---|---|---|
| 1 | Pledge amount | 9223.37 | 9223.37 | 1 | | | | | | | | | | | |
| 2 | Number of Backers | 187.81 | 1010.76 | 0.65 ** | 1 | | | | | | | | | | |
| 3 | Success | 0.26 | 0.44 | 0.25 ** | 0.26 ** | 1 | | | | | | | | | |
| 4 | EO autonomy | 0.11 | 0.26 | 0.15 ** | 0.26 ** | 0.07 ** | 1 | | | | | | | | |
| 5 | EO competitive aggressiveness | 0.12 | 0.30 | 0.00 | −0.01 | 0.00 | −0.01 | 1 | | | | | | | |
| 6 | EO innovativeness | 0.46 | 0.58 | 0.00 | −0.02 | −0.01 | −0.03 * | 0.10 ** | 1 | | | | | | |
| 7 | EO proactiveness | 1.18 | 0.91 | −0.04 * | −0.06 ** | −0.07 ** | −0.05 ** | 0.09 ** | 0.16 ** | 1 | | | | | |
| 8 | EO risk-taking | 0.08 | 0.21 | 0.15 ** | 0.25 ** | 0.07 ** | 0.28 ** | 0.02 | −0.02 | −0.05 ** | 1 | | | | |
| 9 | Entrepreneur gender | 0.13 | 0.34 | 0.03 | 0.07 ** | 0.13 ** | 0.12 ** | 0.09 ** | 0.09 ** | 0.16 ** | 0.15 ** | 1 | | | |
| 10 | Facebook connection | 0.36 | 0.48 | 0.01 | 0.02 | −0.01 | −0.02 | −0.02 | 0.03 | 0.00 | −0.02 | −0.05 ** | 1 | | |
| 11 | External websites | 0.74 | 0.44 | 0.07 ** | 0.07 ** | 0.13 ** | −0.01 | 0.00 | 0.02 | 0.01 | −0.01 | −0.01 | 0.09 ** | 1 | |
| 12 | Video | 0.81 | 0.39 | 0.08 ** | 0.06 ** | 0.08 ** | 0.00 | 0.01 | 0.05 ** | −0.01 | 0.01 | −0.01 | 0.05 ** | 0.22 ** | 1 |
| 13 | ProjectWeLove | 0.10 | 0.29 | 0.24 ** | 0.21 ** | 0.32 ** | 0.05 ** | −0.02 | 0.03 * | −0.02 | 0.04 ** | 0.05 ** | 0.02 | 0.13 ** | 0.13 ** |
| 14 | Duration | 35.90 | 12.92 | 0.01 | −0.02 | −0.11 ** | 0.02 | 0.03 * | −0.01 | 0.00 | −0.01 | 0.00 | −0.07 ** | −0.06 ** | −0.02 |
| 15 | Number of updates | 4.80 | 8.88 | 0.36 ** | 0.33 ** | 0.60 ** | 0.08 ** | −0.03 | 0.00 | −0.06 ** | 0.09 ** | 0.06 ** | 0.06 ** | 0.19 ** | 0.16 ** |
| 16 | Number of comments | 52.35 | 375.21 | 0.82 ** | 0.56 ** | 0.23 ** | 0.16 ** | 0.00 | −0.01 | −0.04 ** | 0.17 ** | 0.03 * | 0.00 | 0.05 ** | 0.06 ** |
| 17 | Number of reward levels | 6.49 | 4.23 | 0.14 ** | 0.14 ** | 0.22 ** | 0.00 | −0.01 | 0.05 ** | −0.02 | 0.03 * | 0.03 * | 0.08 ** | 0.22 ** | 0.29 ** |
| 18 | Number of words | 741.61 | 633.20 | 0.16 ** | 0.12 ** | 0.16 ** | 0.01 | 0.00 | 0.02 | −0.05 ** | 0.03 | −0.02 | 0.04 ** | 0.19 ** | 0.23 ** |
| 19 | Funding goal | 9223.37 | 9223.37 | 0.07 ** | 0.02 | −0.05 ** | −0.01 | 0.00 | 0.00 | 0.00 | 0.02 | −0.01 | −0.01 | −0.02 | −0.01 |
| 20 | Geography: USA | 0.68 | 0.47 | 0.03 * | 0.01 | −0.04 ** | −0.02 | 0.03 * | 0.05 ** | 0.04 ** | −0.01 | 0.02 | 0.08 ** | 0.08 ** | 0.17 ** |
| 21 | Geography: Canada | 0.05 | 0.22 | −0.02 | −0.02 | −0.01 | 0.01 | 0.01 | −0.01 | −0.01 | 0.02 | −0.02 | −0.02 | −0.02 | −0.04 ** |
| 22 | Geography: Latin America | 0.01 | 0.10 | −0.02 | −0.02 | −0.03 * | −0.02 | −0.03 * | −0.05 ** | −0.08 ** | −0.02 | −0.02 | −0.02 | −0.05 ** | −0.06 ** |
| 23 | Geography: Europe | 0.19 | 0.40 | −0.03 * | −0.03 * | −0.03 | −0.01 | −0.05 ** | −0.03 * | −0.02 | −0.02 | −0.04 ** | −0.06 ** | −0.06 ** | −0.16 ** |
| 24 | Geography: Asia | 0.02 | 0.15 | 0.02 | 0.01 | 0.11 ** | 0.01 | 0.03 * | 0.01 | 0.00 | 0.01 | 0.06 ** | −0.06 ** | −0.03 * | 0.03 * |
| 25 | Geography: Oceania | 0.03 | 0.17 | −0.01 | 0.00 | −0.02 | 0.01 | 0.02 | 0.00 | 0.02 | 0.00 | 0.00 | 0.01 | −0.02 | −0.05 ** |

| | | 13 | 14 | 15 | 16 | 17 | 18 | 19 | 20 | 21 | 22 | 23 | 24 | 25 |
|---|---|---|---|---|---|---|---|---|---|---|---|---|---|---|
| 13 | ProjectWeLove | 1 | | | | | | | | | | | | |
| 14 | Duration | −0.04 ** | 1 | | | | | | | | | | | |
| 15 | Number of updates | 0.34 ** | −0.06 ** | 1 | | | | | | | | | | |
| 16 | Number of comments | 0.14 ** | 0.00 | 0.36 ** | 1 | | | | | | | | | |
| 17 | Number of reward levels | 0.23 ** | −0.07 ** | 0.33 ** | 0.11 ** | 1 | | | | | | | | |
| 18 | Number of words | 0.17 ** | −0.03 * | 0.29 ** | 0.12 ** | 0.39 ** | 1 | | | | | | | |
| 19 | Funding goal | −0.02 | 0.05 ** | −0.01 | 0.00 | −0.03 * | −0.01 | 1 | | | | | | |
| 20 | Geography: USA | 0.03 * | −0.03 | 0.02 | 0.02 | 0.11 ** | 0.02 | 0.00 | 1 | | | | | |
| 21 | Geography: Canada | −0.01 | 0.03 | −0.01 | −0.01 | −0.03 | 0.01 | 0.01 | −0.33 ** | 1 | | | | |
| 22 | Geography: Latin America | −0.03 | 0.03 * | −0.04 ** | −0.01 | −0.06 ** | −0.04 ** | −0.01 | −0.15 ** | −0.02 | 1 | | | |
| 23 | Geography: Europe | −0.03 * | 0.01 | −0.06 ** | −0.03 * | −0.10 ** | −0.02 | 0.01 | −0.71 ** | −0.11 ** | −0.05 ** | 1 | | |
| 24 | Geography: Asia | −0.02 * | 0.01 | 0.02 | 0.02 | −0.03 | 0.00 | −0.01 | −0.22 ** | −0.03 * | −0.02 | −0.07 ** | 1 | |
| 25 | Geography: Oceania | −0.03 * | 0.00 | −0.02 | 0.02 | −0.03 * | −0.01 | 0.00 | −0.25 ** | −0.04 ** | −0.02 | −0.08 ** | −0.03 | 1 |

Note. N = 5105. Entrepreneur gender, 0 = Man, 1 = Woman. Facebook connection, 0 = No, 1 = Yes. External website, 0 = No, 1 = Yes. Video, 0 = No, 1 = Yes. ProjectWeLove, 0 = No, 1 = Yes. * $p < 0.05$, ** $p < 0.01$.

Table 4 reports: (a) a model including only control variables; (b) a model including only direct effects for testing of H1; and (c) a model including interaction effects for testing H2a–H2e for all three types of regression models. The findings reported in columns 1b, 2b, and 3b indicated that women entrepreneurs were positively correlated to pledge amount (logged; $b = 0.10$, $p < 0.05$), number of backers ($b = 0.41$, $p < 0.001$) and crowdfunding success ($b = 0.95$, $p < 0.001$), across regression specifications, lending support to H1. With regard to the (un-hypothesized) direct effects of the EO variables on crowdfunding performance, we found that autonomy ($b = 0.06$, $p < 0.001$) and risk-taking ($b = 0.23$, $p < 0.001$) were positively related to number of backers, while competitive aggressiveness (b = −0.08, $p < 0.001$), innovativeness ($b = −0.05$, $p < 0.05$) and proactiveness ($b = −0.07$, $p < 0.001$) were negatively related to number of backers for the negative binomial regression model. The five EO variables were not significantly associated with pledge amount (logged) and crowdfunding success when the models that included only direct effects (b) were considered.

**Table 4.** Regression coefficient for models 1 (OLS), 2 (Negative binomial model) and 3 (Binary logistic model) (dependent variable: crowdfunding performance).

| | Model (1): OLS | | | Model (2): Negative Binomial Model | | | Model (3): Binary Logistic Model | | |
|---|---|---|---|---|---|---|---|---|---|
| | Pledge Amount (Logged) | | | Number of Backers (Logged) | | | Success | | |
| | (a) Control Only | (b) Direct Effect | (c) Full Model | (a) Control Only | (b) Direct Effect | (c) Full Model | (a) Control Only | (b) Direct Effect | (c) Full Model |
| Constant | 2.17 (0.13) *** | 2.12 (0.13) *** | 2.06 (0.13) *** | 5.23 (0.13) *** | 4.04 (0.13) *** | 3.46 (0.14) *** | 19.49 (3531.84) | 19.28 (3532.36) | 19.05 (3464.56) |
| Facebook connection | −0.05 (0.03) | −0.05 (0.03) | −0.05 (0.03) | −0.17 (0.03) *** | −0.04 (0.03) | −0.04 (0.03) | −0.48 (0.12) *** | −0.44 (0.12) *** | −0.42 (0.12) *** |
| External websites | 0.27 (0.03) *** | 0.28 (0.03) *** | 0.27 (0.03) *** | 0.43 (0.03) *** | 0.23 (0.03) *** | 0.22 (0.03) *** | 0.20 (0.13) | 0.23 (0.14) | 0.23 (0.14) |
| Video | 0.69 (0.04) *** | 0.69 (0.04) *** | 0.68 (0.04) *** | 0.12 (0.04) * | 0.49 (0.04) *** | 0.54 (0.04) *** | −0.21 (0.15) | −0.22 (0.15) | −0.25 (0.15) |
| ProjectWeLove | 0.54 (0.05) *** | 0.54 (0.05) *** | 0.54 (0.05) *** | 0.75 (0.05) *** | 0.65 (0.05) *** | 0.65 (0.05) *** | 1.00 (0.18) *** | 1.00 (0.18) *** | 1.05 (0.18) *** |
| Duration | −0.05 (0.01) *** | −0.05 (0.01) *** | −0.05 (0.01) *** | −0.17 (0.02) *** | −0.08 (0.02) *** | −0.08 (0.02) *** | −0.09 (0.06) | −0.09 (0.06) | −0.09 (0.06) |
| Number of updates | 0.43 (0.02) *** | 0.43 (0.02) *** | 0.42 (0.02) *** | 0.96 (0.03) *** | 1.02 (0.03) *** | 1.01 (0.03) *** | 2.76 (0.13) *** | 2.79 (0.13) *** | 2.76 (0.13) *** |
| Number of comments | 0.08 (0.02) *** | 0.08 (0.02) *** | 0.07 (0.02) *** | 1.09 (0.06) *** | 1.04 (0.05) *** | 1.02 (0.05) *** | 9.61 (0.95) *** | 9.75 (0.96) *** | 9.72 (0.97) *** |
| Number of reward levels | 0.26 (0.02) *** | 0.26 (0.02) *** | 0.26 (0.02) *** | 0.24 (0.02) *** | 0.29 (0.02) *** | 0.28 (0.02) *** | 0.10 (0.07) | 0.10 (0.07) | 0.11 (0.07) |
| Number of words | 0.17 (0.02) *** | 0.17 (0.02) *** | 0.17 (0.02) *** | 0.22 (0.02) *** | 0.14 (0.02) *** | 0.15 (0.02) *** | −0.01 (0.07) | 0.00 (0.07) | −0.01 (0.07) |
| Funding goal (logged) | 0.03 (0.02) | 0.03 (0.02) | 0.03 (0.02) | −0.13 (0.01) *** | 0.07 (0.01) *** | 0.07 (0.02) *** | −1.34 (0.07) *** | −1.35 (0.07) | −1.37 (0.07) |
| Geography: USA | −0.05 (0.12) | −0.02 (0.12) | 0.05 (0.12) | −1.64 (0.13) *** | −0.83 (0.13) *** | −0.34 (0.13) * | −20.07 (3531.84) | −20.06 (3532.36) | −19.83 (3464.56) |
| Geography: Canada | −0.26 (0.14) | −0.23 (0.14) | −0.15 (0.14) | −2.20 (0.14) *** | −1.22 (0.15) *** | −0.68 (0.15) *** | −19.99 (3531.84) | −19.93 (3532.36) | −19.72 (3464.56) |
| Geography: Latin America | −0.95 (0.18) *** | −0.93 (0.18) *** | −0.84 (0.18) *** | −2.71 (0.20) *** | −1.68 (0.20) *** | −1.15 (0.20) *** | −19.89 (3531.84) | −19.87 (3532.36) | −19.58 (3464.56) |
| Geography: Europe | −0.21 (0.13) | −0.18 (0.13) | −0.11 (0.13) | −1.84 (0.13) *** | −1.14 (0.13) *** | −0.63 (0.14) *** | −19.80 (3531.84) | −19.74 (3532.36) | −19.46 (3464.56) |
| Geography: Asia | 0.50 (0.15) *** | 0.52 (0.15) *** | 0.58 (0.15) *** | −1.27 (0.16) *** | −0.36 (0.16) *** | 0.18 (0.16) | −18.72 (3531.84) | −18.80 (3532.36) | −18.52 (3464.56) |
| Geography: Oceania | −0.17 (0.15) | −0.13 (0.15) | −0.05 (0.15) | −1.83 (0.15) *** | −0.88 (0.16) *** | −0.36 (0.16) * | −20.05 (3531.84) | −19.99 (3532.36) | −19.75 (3464.56) |
| Entrepreneur gender | | **0.10 (0.04) *** | 0.14 (0.04) *** | | **0.41 (0.05) *** | 0.36 (0.05) *** | | **0.95 (0.14) *** | 1.08 (0.15) *** |
| EO autonomy | | 0.02 (0.01) | −0.01 (0.02) | | 0.06 (0.02) *** | −0.07 (0.02) *** | | 0.11 (0.07) | 0.01 (0.08) |

**Table 4.** *Cont.*

| | Model (1): OLS | | | Model (2): Negative Binomial Model | | | Model (3): Binary Logistic Model | | |
|---|---|---|---|---|---|---|---|---|---|
| | Pledge Amount (Logged) | | | Number of Backers (Logged) | | | Success | | |
| | (a) Control Only | (b) Direct Effect | (c) Full Model | (a) Control Only | (b) Direct Effect | (c) Full Model | (a) Control Only | (b) Direct Effect | (c) Full Model |
| EO competitive aggressiveness | | −0.02 (0.01) | 0.00 (0.02) | | −0.08 (0.01) *** | −0.04 (0.02) * | | 0.07 (0.05) | 0.12 (0.06) * |
| EO innovativeness | | 0.01 (0.01) | 0.03 (0.02) | | −0.05 (0.02) * | −0.03 (0.02) | | −0.05 (0.06) | −0.04 (0.07) |
| EO proactiveness | | −0.03 (0.01) | 0.01 (0.02) | | −0.07 (0.01) *** | 0.00 (0.02) | | −0.09 (0.06) | 0.09 (0.05) |
| EO risk-taking | | 0.00 (0.02) | −0.05 (0.02) * | | 0.23 (0.02) *** | −0.06 (0.02) *** | | −0.10 (0.07) | −0.09 (0.08) |
| EO autonomy × Entrepreneur gender | | | 0.03 (0.03) | | | **0.12 (0.04) *** | | | **0.73 (0.20) *** |
| EO competitive aggressiveness × Entrepreneur gender | | | −0.03 (0.03) | | | −0.04 (0.03) | | | −0.03 (0.12) |
| EO innovativeness × Entrepreneur gender | | | −0.03 (0.03) | | | −0.01 (0.04) | | | 0.11 (0.12) |
| EO proactiveness × Entrepreneur gender | | | **−0.16 (0.04) *** | | | **−0.36 (0.04) *** | | | **−0.94 (0.16) *** |
| EO risk-taking × Entrepreneur gender | | | **0.09 (0.04) *** | | | **0.50 (0.04) *** | | | −0.06 (0.15) |
| R² | 0.494 | 0.495 | 0.5 | | | | 0.721 | 0.728 | 0.737 |
| AIC | | | | 50,758.846 | 49,298.624 | 48,886.420 | | | |
| BIC | | | | 50,869.975 | 49,448.975 | 49,069.456 | | | |

Note. N = 5105. Entrepreneur gender, 0 = Man, 1 = Woman. Facebook connection, 0 = No, 1 = Yes. External website, 0 = No, 1 = Yes. Video, 0 = No, 1 = Yes. ProjectWeLove, 0 = No, 1 = Yes. Standard errors in parentheses. The mean VIF ranges from 2.79 to 3.44. * $p < 0.05$. *** $p < 0.001$. All independent and moderator variables are standardized. In addition, all controls were standardized, except for binary variables. Hypothesized significant coefficients are bolded.

As columns 1c, 2c and 3c in Table 4 indicate, the interactions between signals of autonomy and women entrepreneurs were significantly and positively related to number of backers ($b = 0.12$, $p < 0.05$) and crowdfunding success ($b = 0.73$, $p < 0.001$) in the regression model 2 and 3, lending partial support to H2a. However, the interactions of competitive aggressiveness and women entrepreneurs were not significantly related to crowdfunding performance in any of the relevant regression models, which led to the rejection of H2b. Similarly, H2c was also rejected, as there were no significant relationships between the interaction of innovativeness and women entrepreneurs. The interactions between signals of proactiveness and women entrepreneurs were significantly and negatively related to crowdfunding performance in all regression models, with significance levels of $p < 0.001$ (with coefficients ranging between −0.16 and −0.94), rejecting H2d. The interactions between signals of risk-taking and women entrepreneurs were significantly and positively related to pledge amount ($b = 0.09$, $p < 0.05$) and number of backers ($b = 0.50$, $p < 0.001$) in the regression model 1 and 2, giving support to H2e. Support for H2e was mixed, since only some regression coefficients were significant and positive. Table 5 summarizes the results of the hypotheses testing.

Figures 2–4 illustrate the moderating effects of autonomy, risk-taking, and proactiveness signals in the three models. Slopes using autonomy as a moderator indicated that the slopes between women entrepreneurs and crowdfunding performance were relatively flat, except when there were high signals of autonomy; specifically, for Figure 2a, simple slope tests indicated that women entrepreneurs were positively related to number of backers (0.53; $p < 0.001$) when autonomy signals were high (1 SD above the mean) and positively related to number of backers (0.41; $p < 0.001$) when autonomy signals were low (1 SD below the mean). For Figure 2b, simple slope tests indicated that women entrepreneurs were positively related to crowdfunding success (1.68; $p < 0.001$) when autonomy signals were

high (1 SD above the mean), and positively related to crowdfunding success (0.95; $p < 0.001$) when autonomy signals were low (1 SD below the mean).

**Table 5.** Results of the hypothesis tests.

| Hypothesis | Results |
|---|---|
| H1: In reward-based crowdfunding, women entrepreneurs are more likely than men to obtain financial backing. | Supported |
| H2a: The relationship between a woman entrepreneur and reward-based crowdfunding performance is stronger when the entrepreneur has a high degree of autonomy than when she does not. | Partially Supported |
| H2b: The relationship between a woman entrepreneur and reward-based crowdfunding performance is stronger when the entrepreneur has a high degree of competitive aggressiveness than when she does not. | Not Supported Contrary to the null (Insignificant) |
| H2c: The relationship between a woman entrepreneur and reward-based crowdfunding performance is stronger when the entrepreneur has a high degree of innovation than when she does not. | Not Supported Contrary to the null (Insignificant) |
| H2d: The relationship between a woman entrepreneur and reward-based crowdfunding performance is stronger when the entrepreneur is highly proactive than when she is not. | Not Supported Contrary to the null (Significant) |
| H2e: The relationship between a woman entrepreneur and reward-based crowdfunding performance is stronger when the entrepreneur is highly risk-taking than when she is not. | Partially Supported |

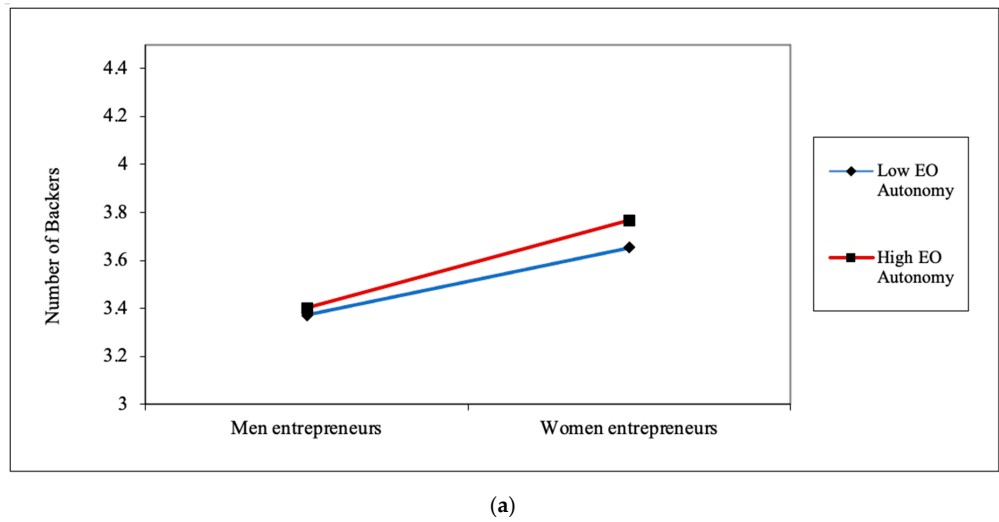

(**a**)

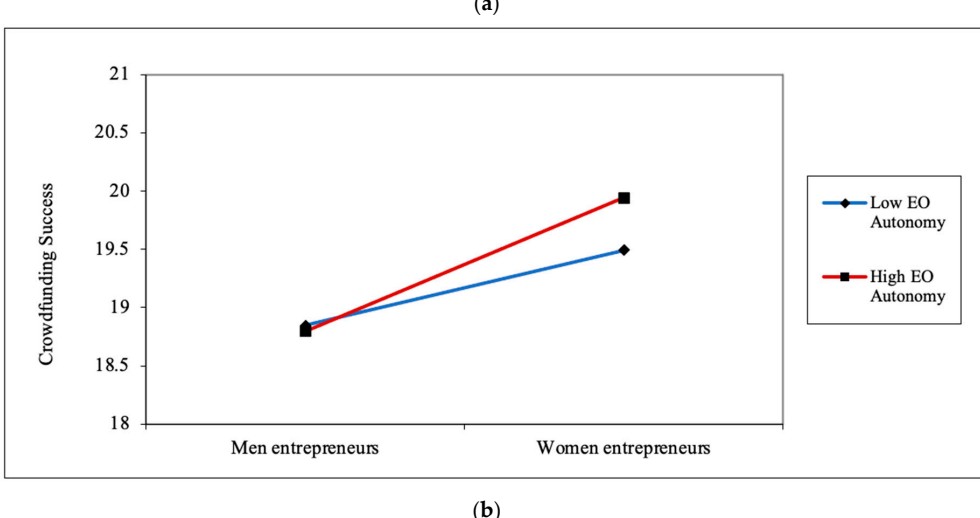

(**b**)

**Figure 2.** Simple slope analysis of the EO 'autonomy's moderating effect on the relationship between women entrepreneurs and crowdfunding performance, based on models 2 and 3. In all figures, "low" refers to 1 SD below the mean, and "high" refers to 1 SD above the mean: (**a**) Moderate effect of the EO of 'autonomy' in the backers model; (**b**) Moderate effect of the EO of 'autonomy' in the status model.

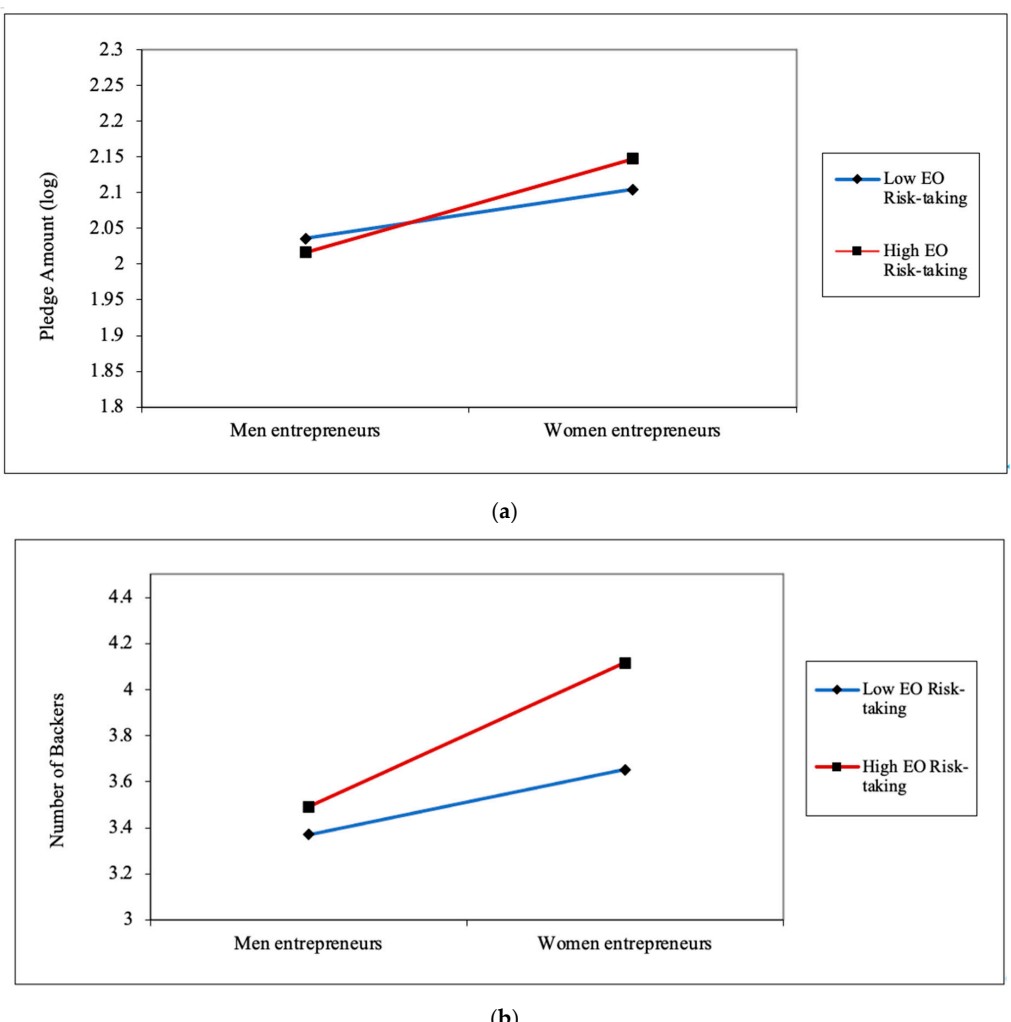

(a)

(b)

**Figure 3.** Simple slope analysis of the EO of 'risk-taking's moderating effect on the relationship between women entrepreneurs and crowdfunding performance, based on models 1, 2, and 3. In all figures, "low" refers to 1 SD below the mean, and "high" refers to 1 SD above the mean: (**a**) Moderate effect of the EO of 'risk-taking' in pledge amount model; (**b**) Moderate effect of the EO of 'risk-taking' in backers model.

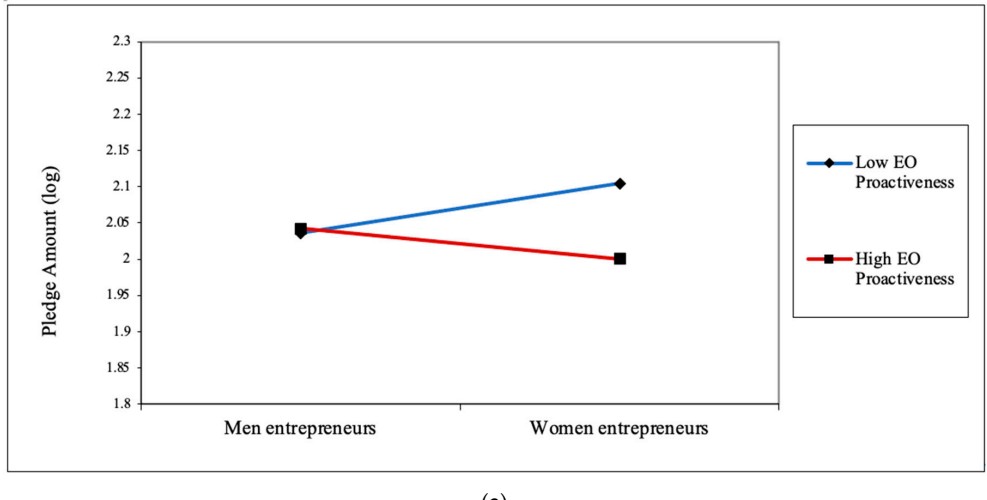

(a)

**Figure 4.** *Cont.*

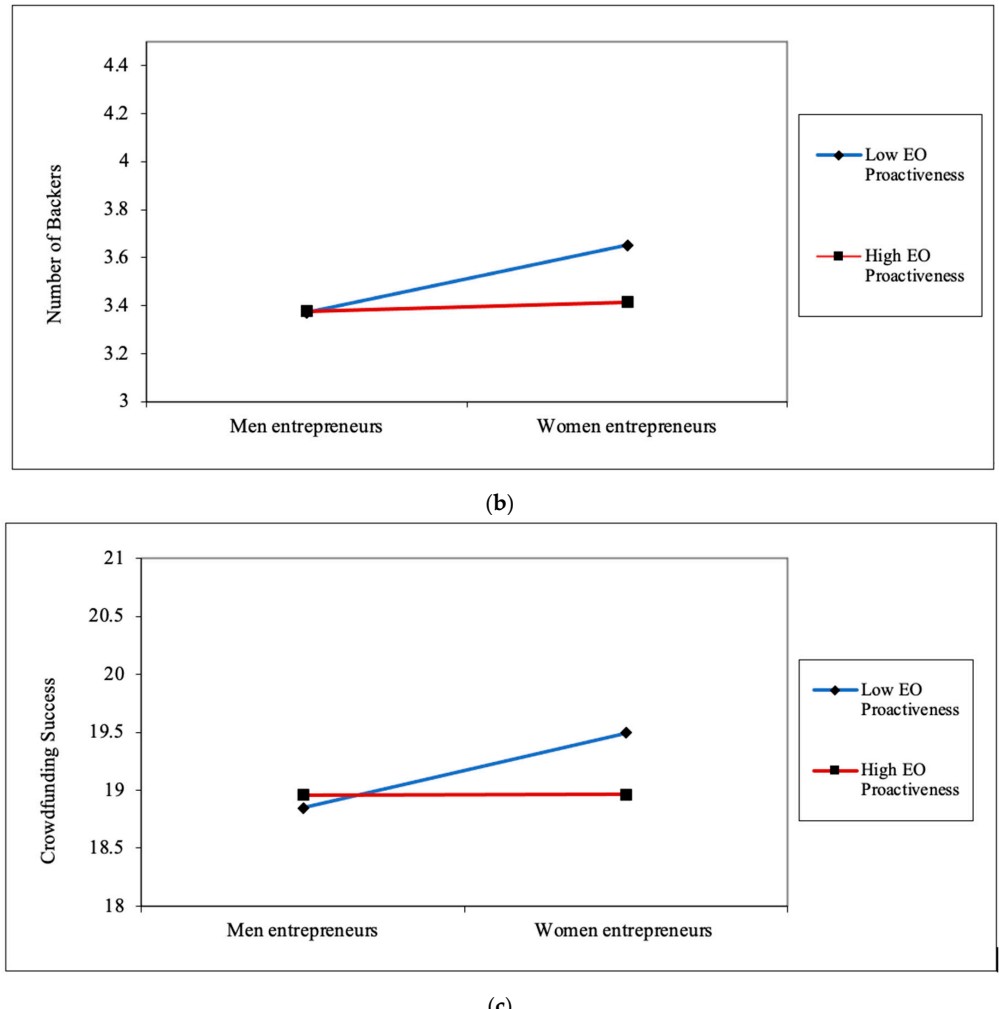

**Figure 4.** Simple slope analysis of the EO of 'proactiveness's moderating effect on the relationship between women entrepreneurs and crowdfunding performance, based on models 1, 2, and 3. In all figures, "low" refers to 1 SD below the mean and "high" refers to 1 SD above the mean. (**a**) Moderate effect the of EO of 'proactiveness' in the pledge amount model; (**b**) Moderate effect of the EO of 'proactiveness' in the backers model. (**c**) Moderate effect of the EO of 'proactiveness' in the status model.

Slopes using risk-taking as a moderator indicated that the slopes between women entrepreneurs and crowdfunding performance were relatively flat, except when there were high signals of risk-taking; specifically, for Figure 3a, simple slope tests indicated that women entrepreneurs were positively related to pledge amount (0.19; $p < 0.05$) when risk-taking signals were high (1 SD above the mean), and positively related to pledge amount (0.10; $p < 0.05$) when risk-taking signals were low (1 SD below the mean). For Figure 3b, simple slope tests indicated that women entrepreneurs were positively related to the number of backers (0.92; $p < 0.001$) when risk-taking signals were high (1 SD above the mean), and positively related to pledge amount (0.41; $p < 0.001$) when risk-taking signals were low (1 SD below the mean).

Using proactiveness as a moderator, simple slope analysis generally indicated that the relationships between women entrepreneurs and crowdfunding performance were rather flat, except when there were low signals of proactiveness; specifically, for Figure 4a, simple slope tests indicated that women entrepreneurs had a positive effect on pledge amount (0.10; $p < 0.05$) when proactiveness signals were low (one SD below the mean), whereas the relation between women entrepreneurs and pledge amount was not significant (−0.06;

$p > 0.05$) when proactiveness signals were high (one SD above the mean). For Figure 4b, simple slope tests indicated that women entrepreneurs had a positive effect on the number of backers (0.41; $p < 0.001$) when proactiveness signals were low (one SD below the mean), whereas the relation between women entrepreneurs and the number of backers was not significant (0.06; $p > 0.05$) when proactiveness signals were high (one SD above the mean). For Figure 4c, simple slope tests indicated that women entrepreneurs had a positive effect on crowdfunding success when proactiveness signals were low (0.95; $p < 0.001$), whereas it was not significant when proactiveness signals were high (0.01; $p > 0.05$). The simple slope tests were very similar for the other models.

In practical terms, for instance, H2a which was partially supported in the model 2 and 3, implying that a women entrepreneur who had high signals of autonomy was more likely to be recognized by crowd funders, and thus to succeed. However, having high signals of autonomy did not help women entrepreneurs to increase backers' pledge amounts. Although leading to a rejection of H2d, signals of proactiveness had a moderate effect on the relationship between women entrepreneurs and crowdfunding performance. Women entrepreneurs with a high degree of proactiveness were more likely to decrease backers' pledge amount as well as number of backers, which made it harder to succeed in crowdfunding. As for the moderate effect of the EO dimension of risk-taking, this was also partially supported in the model 1 and 2; this implied that a woman entrepreneur who had a high degree of risk-taking was more likely to attract investors and raise their pledge amount, but the improvement effect was not strong enough to turn a campaign from failure to success, reflecting the limitations of risk-taking signals in a crowdfunding context.

## 5. Discussion and Contributions

Based upon previous content-centric research into reward-based crowdfunding and business fundraising, our research results revealed that EO signals were significant indicators of crowdfunding performance for women entrepreneurs; to be specific, signals of autonomy and risk-taking positively moderated the relationship between women entrepreneurs and crowdfunding performance, while signals of proactiveness were a negative moderator of this relationship. In addition, signals of innovativeness and competitive aggressiveness had no moderate effect on that relationship. Women entrepreneurs were compensated more for releasing high signals of autonomy and risk-taking, supported by the literature regarding social identity theory [21,36]. Backers were in favor of women entrepreneurs who showed strong signals of autonomy and risk-taking, and who were able to transform existing organizations by updating strategic capabilities [75] and who viewed uncertainties as "opportunities".

Even though we expected signals of proactiveness to be significant for women entrepreneurs to achieve crowdfunding success, we were nonetheless surprised to learn from our regression analysis that they had a moderately negative influence on the success of female entrepreneurs' campaigns for reward-based crowdfunding. This result may be explained by the fact that reward-based crowdfunding requires less proactiveness and initiative from women entrepreneurs. In other words, being influenced by gender stereotypes [81,82], crowd funders might prefer women entrepreneurs to be less proactive, and which would be in line with the backer's expectations of gender roles. Lastly, the performance of female entrepreneurs in obtaining a fund was not significantly impacted by signals of innovativeness and competitive aggressiveness. This may be due to the fact that the mission of such reward-based crowdfunding platforms as kickstarter.com primarily aims 'to help bring creative projects to life' (See https://www.kickstarter.com/charter), and especially so in the technology and design industry, where most women entrepreneurs pursue innovation.

### 5.1. Theoretical Contributions

We contribute to the research in three major ways: First, this study adds to the expanding body of knowledge on reward-based crowdfunding campaigns and female

entrepreneurs. We sought to find the driving EO signals behind the funding advantages for women entrepreneurs in reward-based crowdfunding, based upon social identity theory. Although prior studies have suggested that women entrepreneurs have certain advantages in online crowdfunding, little is known about the contribution of EO signals for the gender gap [26]. As this study demonstrates, in reward-based crowdfunding, those women entrepreneurs who presented high signals of autonomy and risk-taking were rewarded, whereas releasing strong signals of proactiveness counted against their success. This study hoped to enable an in-depth understanding of the link between investors' decision-making and women's entrepreneurial behaviors [27,28], in addition to testing whether the social identity concept-derived EO mechanisms were a source of advantage for women entrepreneurs.

Second, this study contributed to the social identity literature by exploring entrepreneurs' behavioral orientations in crowdfunding, responding to the call of gender scholars to examine heterogeneity among women and men entrepreneurs [29,30]. From the perspective of entrepreneurial behavioral orientations, our study not only went beyond demographics to reveal the social psychological mechanism that affects the judgment of investors, but also investigated the five dimensions of EO of different genders in a reward-based crowdfunding context [31]. In this regard, we took the lead in applying the social identity theory in start-up financing and concentrated on female entrepreneurs. We contend that, in contrast to male entrepreneurs, the underlying social psychological mechanism of their potential backers are less well-known. Women entrepreneurs who want to have successful campaigns and draw in potential investors face additional obstacles in their entrepreneurial behaviors as a result of such confused expectations. This study implied that women entrepreneurs may be able to overcome these barriers to entrepreneurial behavior by turning to specific signals of entrepreneurial orientation that are align with crowd funders' expectation. In the setting of reward-based crowdfunding, women entrepreneurs should increase the usage of autonomy orientation words such as "autonomous", "empowered", "self-directed" and "unblock", as well as risk-taking orientation words such as "adventurous", "courageous", "speculative" and "unsettled" in their crowdfunding pitches; at the same time, they should reduce the usage of proactiveness orientation words, for instance, "ambitious", "explorative", "formulate" and "opportunistic".

Third, while prior studies have emphasized critical issues surrounding financing of women-led ventures [20], there has been little theory-driven research on this subject. Here, our analysis offers two crucial findings for further investigation. First, as the factors influencing backers' judgement in the two situations could be different, it is not always easy to simply adapt common entrepreneurial financial notions from the funding of men-led microenterprises into the setting of women-led microenterprises. Second, the importance of entrepreneurial behavior is then illustrated in relation to women entrepreneurs, a situation where the stakeholders' expectations are ambiguous. We highlight that women entrepreneurs have to move beyond trust to attract these potential investors who take advantage of the entrepreneurial orientation of 'autonomy' and 'risk-taking'.

### 5.2. Practical Implications

The research indicates that specific entrepreneurial behavior is important for women entrepreneurs from a practical perspective. In addition to being more trusted by backers due to gender stereotypes and information asymmetry in reward-based crowdfunding [20], women entrepreneurs should be aware of their entrepreneurial tendencies while turning to the crowd for resources. Although women entrepreneurs have certain advantages in reward-based crowdfunding, it is particularly important for them to release more signals of autonomy as well as risk-taking and fewer signals of proactiveness in their campaign narratives. Women entrepreneurs, particularly those in their early stages and many of whom seek crowd funding, struggle to become self-reliant and audacious owning to being influenced by gender bias [17]. This study is yet another reminder of the prominence of having the right entrepreneurial behavior from the beginning on. Furthermore, even

though proactive firms are viewed as market leaders and often signal a more competitive and successful venture, it is important for women entrepreneurs to carefully display high proactiveness signals in their campaign pitches, since excessive proactiveness is not consistent with backers' expectations of women entrepreneurs.

### 5.3. Avenues for Future Research

Notwithstanding its achievements, this research leaves us with certain unsolved problems that need to be resolved in future. First, microbusiness owners frequently use crowdfunding platforms, but most backers on crowdfunding platforms are individual investors without professional investment experience. Therefore, a promising direction for future research would be to examine the susceptibility of traditional equity investors to entrepreneurial behaviors. Second, as in earlier research on reward-based crowdfunding performance, this study did not survey funders directly but only analyzed the aggregate funding decisions documented in the online crowdfunding context, and drew conclusions based on social identity theory [21,36] and gender bias [17]. Future studies should consider surveying crowd funders themselves to resolve this limitation and determine further factors affecting crowdfunding decisions.

Third, we were astonished to learn that signals of innovativeness and competitive aggressiveness do not predict the success of reward-based crowdfunding initiatives run by women. In a similar line, in contrast to certain research studies undertaken in the reward-based crowdfunding environment [111], this research did not discover a connection between the success of their crowdfunding campaigns and the linguistic preferences of female business owners. Therefore, a feasible direction for future research would be to find other pertinent linguistic styles and evaluate whether their use helps women-led enterprises more than men-led enterprises, also on other types of crowdfunding, including equity crowdfunding and loan-based crowdfunding platforms.

Fourth, by adopting a "word count approach" to evaluate entrepreneurs' EO [111], another drawback of the study was that this method may have overlooked some subtleties of complex scenarios, such as the measurement of entrepreneurial behaviors. Notwithstanding its efficacy and accuracy in processing large quantities of data, we were only able to make judgements about how the startups portrayed themselves on reward-based crowdfunding platforms; we were unable to draw conclusions about the startups' genuine entrepreneurial approach. Therefore, we recommend that future studies investigate the actual entrepreneurial behaviors of start-ups and other interesting aspects of entrepreneurial behaviors to infer entrepreneurs' EO and theorize on the differences between words and behaviors.

**Author Contributions:** Conceptualization. K.Z. and H.W.; methodology, K.Z.; software, K.Z. and W.W.; validation, K.Z. and W.W.; writing—original draft preparation, K.Z.; writing—review and editing K.Z. and W.W.; visualization, W.W.; supervision, H.W.; project administration, K.Z.; funding acquisition, H.W. All authors have read and agreed to the published version of the manuscript.

**Funding:** This work was supported by the National Natural Science Foundation of China [71771177, 72072062], and 2022 Huaqiao University Graduate Education and Teaching Reform Research Funding Project (22YJG005).

**Institutional Review Board Statement:** Not applicable.

**Informed Consent Statement:** Not applicable.

**Data Availability Statement:** The data supporting this study's findings are available from the corresponding author upon reasonable request.

**Conflicts of Interest:** They have no conflict of interest or commercial interest to disclose.

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
