# Peer review of "Women Entrepreneurs Who Break through in Reward-Based Crowdfunding: The Influence of Entrepreneurial Orientation"

_sustainability, doi:10.3390/su15129276_

Round 1

Reviewer 1 Report

The article seems interesting. It deals with an important topic worth analyzing. It has, however, a major flaw that deters its publication: the model has gender (sex) as an antecedent when it is hard to imagine that this can be an explanatory variable. Moreover, they use the concept of EO, but did not test it. They interpretatively analyzed text and categorized it according to the interpretation they have. This is hardly Entrepreneurial Orientation.

Reviewer 2 Report

The theoretical part of the paper could be improved by involving a few articles dealing with the gender issue in crowdfunding.

The paper “Ullah, S., & Zhou, Y. (2020). Gender, Anonymity and Team: What Determines Crowdfunding Success on Kickstarter. Journal of Risk and Financial Management, 13(4), 80. DOI.3390/jrfm13040080” is exploring the gender issue on Kickstarter, as the paper does.

However, regarding gender issues on crowdfunding platforms, huge attention has been given to equity crowdfunding. I believe the paper could be improved by mentioning the results of these studies and explaining the differences between reward-based crowdfunding and equity crowdfunding regarding the gender issue. Here are some studies on the gender issue at the equity crowdfunding platforms.

Battaglia, F., Manganiello, M., & Ricci, O. (2021). Is Equity Crowdfunding the Land of Promise for Female Entrepreneurship?. PuntOorg International Journal 6(1), 12-36.

Malaga, R., Mamonov, S., & Rosenblum, J. (2018). Gender difference in equity crowdfunding: an exploratory analysis. International Journal of Gender and Entrepreneurship, 10(4), 332-343. DOI.10.1108/IJGE-03-2018-0020

Mohammadi, A., & Shafi, K. (2018). Gender differences in the contribution patterns of equity-crowdfunding investors. Small Business Economics, 50(2), 275-287. DOI.10.1007/s11187-016-9825-7

Reviewer 3 Report

Thank you for your invitation. The manuscript has a novel topic and a sufficient amount of work. Some modifications will improve the quality of the manuscript.

The content of the abstract is too simple and does not clearly express the theme, research methods, and contributions of the paper.

The introduction lacks an introduction to the research methods and does not clearly state the structure of the paper.

The position of Fig.1 Overview of research model seems to be incorrect.

The logic of the hypotheses is unclear. For example, in H1A, is there a conceptual difference between reward-based crowdfunding and crowdfunding? The reason for proposing it this way is not clearly explained.

In section 2.4. The moderating effects of the Entrepreneur Orientation, Autonomy being on a separate line is quite abrupt. The manuscript still needs to be organized according to the journal's format.

In section 3.1, "Online crowdfunding offers individual investors an opportunity to provide microloans to microbusinesses worldwide through online platforms." This sentence seems to introduce a specific method of crowdfunding, which should have some background. The manuscript should first explain how many types of crowdfunding there are and why this type was chosen for the study. The subsequent content of the manuscript should also be arranged based on the research design. It is recommended to reorganize this part.

A brief description of the data collection process for this platform should also be provided, including whether unique permissions are required, which will be helpful for other researchers to reproduce the data.

In the phrase "Women Entrepreneurs will have a value 283," pay attention to spelling errors. It is recommended to have a language edit.

The manuscript can provide a summary table of variables for a more intuitive understanding.

In the sentence "For hypotheses testing, we built three models for different dependent variables," the models are not found in the manuscript.

If the table content is too long, consider providing it as an appendix or reorganizing it.

The software used for data analysis should be mentioned.

It is recommended to add a summary table of hypothesis results.

In section 5. Discussion and contributions, there is a lack of a summary of the research results. The key points of the study's innovation should be emphasized. The language in this part can be reduced from the researcher's perspective, such as “our” and it is recommended to reorganize it.

Round 2

Reviewer 1 Report

I am afraid that the authors did not understand my point. It is clear to me that an explanatory variable helps to explain a dependent variable. In this case, the authors are assuming that gender explains/causes performance. This is totally awkward as seems to me ill-defined. Gender is a category and performance could be analyzed based on a simple comparison between the two categories. If you find the differences across gender, then you need to explain/find the reasons for explaining the different performances.

Reviewer 3 Report

Some of the tables of the empirical results are not reported in a standard way. Table 2 Regression coefficient for models 1 does not indicate the contents of the parentheses. It is recommended that all tables be organized before publication. Note the brevity of the language.  
